# ITERATIVE CAUSAL DISCOVERY WITH ACTIVE INTERVENTIONS

## ABSTRACT

Multi-entity causal discovery is a fundamental problem in machine learning. Understanding the underlying causal relations is important for counterfactual reasoning and robustness. However, the causal structure is only identifiable up to a *Markov Equivalence Class* with observational data. In real life, it is usually hard or even unethical to gather interventional data. Fortunately, the presence of simulators allows for the production of real or simulated intervention data that can help in identifying causal graphs. We propose a causal discovery algorithm that can iteratively and actively gather intervention data to improve the prediction of causal graphs. We demonstrate on several datasets that iterative interventional data augmentation improves both causal discovery and dynamics prediction performance.

## 1 INTRODUCTION

Multi-entity interactive systems are prevalent in our lives, such as human body joint movements (CMU, 2003), robots navigation, and autonomous driving. Modeling causal relations and predicting forward dynamics among interactive entities have always been a core task in robotics (Li et al., 2020a; Yi et al., 2019; Li et al., 2020b; Kipf et al., 2018; Song et al., 2020). Previous work have designed several physical synthetic datasets for causal relational and structural learning (Yi et al., 2019; You & Han, 2020; Ramanishka et al., 2018; McDuff et al., 2021; Schölkopf et al., 2021; Pearl, 2009). Inferring the causal structure in interactive systems makes model predictions explainable and robust (Smith & Ramamoorthy, 2020). Furthermore, it opens up opportunities for counterfactual data augmentation that is applicable to fields such as reinforcement learning (Pitis et al., 2020).

Generalizing outside the observational data is critical for robust predictions of multi-entity trajectories and causal structures. However, theoretically, given only the observational data, the causal graph is only identifiable up to a Markov equivalent class (Spirtes et al., 2000). Fortunately, interventional data could produce local changes in the conditional distribution and help us identify the true underlying causal graph (Scherrer et al., 2021; Brouillard et al., 2020; Ke et al., 2020). When we are at a traffic scene, we observe the current state and we are capable of simulating other interventional and counterfactual scenarios that could have happened at the moment or in the future. We have the ability to change the current belief of the causal structure through simulating interventions before making predictions. This motivates us to discover means of aggregating interventional data from simulators to improve causal discovery performance. With a simulator, there are a number of intervention targets to choose from. To efficiently utilize the intervention resources, which is often considered costly, (Scherrer et al., 2021) suggests the active selection of intervention targets that can be readily integrated into many differential causal discovery algorithms.

We propose a causal discovery algorithm with an intervention augmentation mechanism that can actively and iteratively choose intervention targets based on the entropy of the posterior. Those targets are intervened to augment the observational dataset for causal discovery and to improve both causal discovery performance and downstream prediction accuracy.

## 2 BACKGROUND AND RELATED WORK

**Causal Structure Learning.** Given samples from some joint distribution, inferring the ground truth causal graph $\mathcal{G}$ is called *causal structure learning* or *causal discovery* (Pearl et al., 2000; Peters

et al., 2017). Most existing works infer the causal graph through observational data. Constraint-based methods usually utilize conditional independence tests to narrow down the potential causal structure graphs that may have produced the data samples (Spirtes et al., 2000), and allow for the presence of latent confounders (Hyttinen et al., 2014; Heinze-Deml et al., 2018). Score-based methods search for all possible directed acyclic graphs (DAGs) and maximize a defined score function $\mathcal{S}(\mathcal{G})$ (Chickering, 2002; Nandy et al., 2018). Continuous-optimization methods reformulate the combinatorial search problem into a continuous optimization problem through a smooth and exact characterization of the acyclic constraint (Zheng et al., 2018a).

Instead of directly searching and evaluating the causal graphs, works such as Neural Relational Inference (NRI) (Kipf et al., 2018), Amortized Causal Discovery (Löwe et al., 2022), and Visual Causal Discovery Network (VCDN) (Li et al., 2020b) use graphical neural networks to simultaneously model the forward dynamics and the causal interactions. This framework uses an encoder to explicitly model the interactive relations as graphs and make trajectory predictions based on the static graph with a decoder. While these models assume the causal relationships remain constant over time, Dynamic-NRI (Graber & Schwing, 2020) and EvolveGraph (Li et al., 2020a) proceed further to dynamically predict the relational graph over time.

**Intervention.** An intervention applied to a target variable $X_i$ in the graph is defined as the local change in the conditional distribution: $p_i\left(X_i \mid \mathrm{pa}\left(X_i\right)\right) \to \tilde{p}\left(X_i \mid \mathrm{pa}\left(X_i\right)\right)$, where $\mathrm{pa}(x)$ denotes the parent nodes of $x$. Previous works have utilized intervention data to improve generalization. For instance, artificial expert intervention has proven to be effective in imitation learning, letting human experts intervene when a region of the state-action space is not *good enough* in autonomous driving (Spencer et al., 2020). The Invariant Causal Prediction (ICP) algorithm (Heinze-Deml et al., 2018) has been proposed to explore the underlying causal structure by exploiting the invariances of the causal relations under different interventions. Peters et al. (2016) also exploits the invariance among causal models in prediction under various intervention setups. Our work aims to apply intervention efficiently in the context of multi-entity dynamics predictions and causal discovery.

## 3  METHODS

Given the temporal trajectory data of multiple entities, we model the relations as a directed acyclic graph $\mathcal{G} = (\mathcal{V}, \mathcal{E})$. The vertices $\mathcal{V}$ contain information about the node attributes and the edge $e_{ij} \sim \mathcal{E}$ contains relational information between node $i$ and node $j$. The node attributes of vertices at time t are denoted as $x_t$. As shown in Figure 1, given observational data $\mathcal{D}_{\mathcal{O}}$ generated from an underlying generative system with fixed or time-varying causal structure and causal parameters, we aim to discover the underlying causal graph $\mathcal{G}$ and make predictions with iteratively and actively collected data from soft interventions. Data gathered from the joint distribution $P$ without interventions are called observational data, and otherwise called interventional data.

**Graph discovery module.** First, given the temporal sequence data $x_{1:T}$, we use the graph discovery module $\mathcal{F}$ to get a posterior distribution on the edges of $\mathcal{G}$. We adopt the same graph discovery architecture as in VCDN (Li et al., 2020b), where a graph neural network is used as a spatial encoder that embeds the nodes and edges for each time frame. The information on the temporal dimension is aggregated with a gated recurrent unit or 1-D convolutional neural network, which supports varying input sizes. Finally, another graph neural network will take in the aggregated information and output a posterior distribution of the causal structure over the edge types: $p\left(\tilde{\mathcal{E}} \mid x_{1:T}\right) \triangleq \mathcal{F}\left(x_{1:T}\right)$. To backpropagate gradients through sampling from this discrete distribution, the Gumbel-Softmax technique (Jang et al., 2016) is used to approximate the gradients.

**Active intervention selection.** To efficiently collect interventional data that will provide more informative knowledge about the causal graph, an intervention policy $I$ will be used to select intervention targets $v_{int}$ that the model is currently least confident about. Based on the entropy of the posterior, it chooses the edges with the lowest confidence about its edge type and set the nodes connected with these edges as targets $v_{int}$, concretely $v_{int}$ are connected by $\arg\min_{e_l} \max_k p\left(e_l == k \mid x_{1:T}\right)$, where $e_l == k$ denotes the event that $l$ th edge is of edge type $k$.

**Iterative interventional data collection.** Suppose the ground truth parameter of the generation system is $W$, given an intervention target node $v_{int}$, we sample a random value $c$ and conduct intervention as $v_{int} \sim \mathrm{do}\left(\mathrm{pa}\left(v_{int}\right) = \mathrm{c} \mid W\right)$, where. Interventional data $d_I = \{\hat{x}_t \ldots \hat{x}_{t+n}\}$ will be

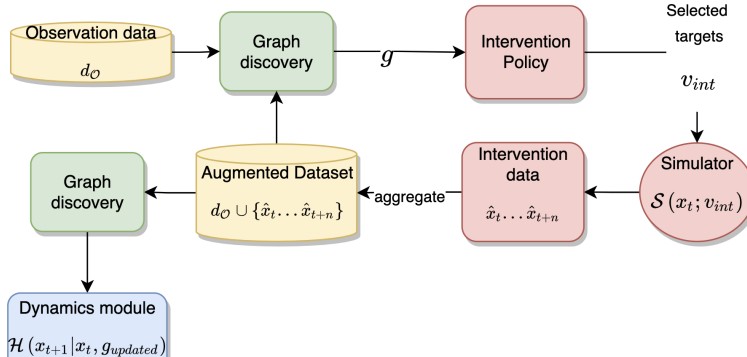

Figure 1: Algorithmic Overview of Iterative Causal Discovery with Active Interventions. During training, interventional data can be actively and iteratively gathered from the simulator and aggregated with the observational data to improve generalization in causal discovery and dynamics prediction tasks.

---

**Algorithm** Iterative Causal Discovery with Active Interventions

---

**Require:**
    Observational data $\mathcal{D}_\mathcal{O}$, Simulator $Sim$
**Initialize:**
    Graph discovery module $\mathcal{F}$, Intervention policy $\mathcal{I}$, Dynamics module $\mathcal{H}$
**while** *not converge* **do**
    Sample a training example $d_O$ from observational data $\mathcal{D}_\mathcal{O}$
    Infer initial causal graph $g \sim \mathcal{F}(d_O)$
    Select intervention targets $v_{int} \sim \mathcal{I}(g)$
    **for** $t = 0; t < T; t{+}{+}$ **do**                     ▷ T roll out steps into the future
        **if** *collect intervention data* **then**
            Intervene on $v_{int}$ and collect $\{\hat{x}_t \ldots \hat{x}_{t+n}\} \sim Sim(x_t, v_{int}, n)$ for $n$ steps
            Infer causal graph with augmented data $g_{updated} \sim \mathcal{F}(d_O \cup \{\hat{x}_t \ldots \hat{x}_{t+n}\})$
            Select intervention targets $v_{int} \sim \mathcal{I}(g_{updated})$
        **end if**
        Predict next step $x_{t+1} \sim \mathcal{H}(x_{t+1} \mid x_t, g_{updated})$       ▷ Exploit current belief of graph
    **end for**
**end while**

---

collected for n steps from the simulator. We augment current sample's observational dataset $d_O$ with the intervention data $d_O \cup \{x_t \ldots x_{t+n}\}$. Then, the graph discovery module will update the current belief of the graph by inferring again with the augmented dataset, producing an updated graph $g_{updated}$. This interventional data augmentation stage can be applied iteratively.

**Dynamics module.** Conditioning on the updated causal graph $g_{updated}$ and current state $x_t$, the dynamics module $\mathcal{H}$ will predict the next step $x_{t+1} \sim \mathcal{H}(x_{t+1} \mid x_t, g_{updated})$ for the nodes in the graph, completing one roll out step of the training pipeline illustrated in the algorithm box above. In implementation, we choose graph recurrent network as the architecture of $\mathcal{H}$.

## 4   Experiments

We evaluate our algorithm on the multi-body interaction environment and synthetic time-varying structural equation model for the performance of causal discovery and prediction task. Additionally, we test the combination of our iterative intervention gathering module and several baseline algorithms on the linear model with Gaussian noise.

### 4.1   Multi-Body Environment

**Data collection and simulator.** We use the Pymunk simulator to spawn 5 balls in a fixed square-shaped 2D space. In each episode, 5 balls will be randomly placed inside the square frame and the relation between each pair of the balls is randomly selected from one of {rigid rod, spring, No relation} with equal probability. We followed the same environment set up as in VCDN (Li

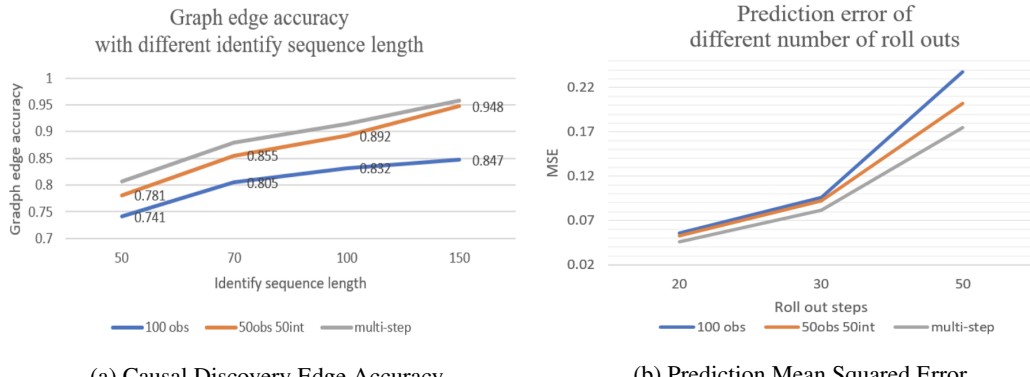

(a) Causal Discovery Edge Accuracy  (b) Prediction Mean Squared Error

Figure 2: Effect of adding interventional data compared to equal sample sized observational data

et al., 2020b) and generated a dataset of 5000 episodes of 500 frames for training and testing. For intervention, after selecting the targets of intervention, we apply intervention to the balls by changing their positions within a circle of predefined radius that satisfies the environment constraints.

**Effect of augmenting interventional data during training.** To test if augmenting interventional data during training helps causal discovery and dynamic prediction, we compared training with only observational data (100 time steps) and with half observational data (50 time steps) and half interventional data (50 time steps) in the multi-body environment. We further compared collecting multiple rounds of interventional data during training, named as "multi-step" in Figure 2. At inference time, we do not assume access to interventional data and we evaluate on various identify sequence length and roll out steps for the three training set up. As shown in Figure 2, the augmented interventional data improves both the identifiability of the causal graph and the prediction performance as compared to equally sized observational data. The improvement continues with more rounds of interventional data collected iteratively and actively.

## 4.2 TIME-VARYING STRUCTURAL EQUATION MODEL DATASET

**Time-varying dataset generation.** We generate a synthetic dataset of varying causal structure with three nodes for T steps: $\mathbf{x}^{(\mathbf{t})} = \{x_1^{(t)}, x_2^{(t)}, x_3^{(t)}\}$, for $t = 1, \ldots, T$. The nodes value over time and the relationship between the nodes are described by the following equation:

$$
\begin{aligned}
x_1^{(t)} &= x_1^{(t-1)} + \alpha^{(t)} x_2^{(t-1)} + \epsilon_1^{(t)} \\
x_2^{(t)} &= x_1^{(t-1)} x_3^{(t-1)} + \epsilon_2^{(t)} \\
x_3^{(t)} &= c + \epsilon_3^{(t)}
\end{aligned}
\quad (1)
\qquad
\alpha^{(t)} = \left\{
\begin{array}{ll}
0 & 1 < t \le 50 \\
1 & 50 < t \le 150 \\
0 & 150 < t \le 200
\end{array}
\right.
\quad (2)
$$

Node values $\{x_1^{(0)}, x_2^{(0)}, x_3^{(0)}\}$ are initialized to 0. The $\epsilon_i^{(t)}, i = 1, 2, 3$ are all Gaussian variables. The constant $c$ is sampled from $-1, 0, 1$ randomly and $\alpha^{(t)}$ varies over time as in equation 2.

We collect a temporal dataset for $T = 200$ steps. To compare causal structure learning without intervention and with intervention, we calculate the True Positive Rate (TPR), True Negative Rate (TNR), F1 Score, and Accuracy of the predicted graph. We tested our algorithm with various rounds of iterative intervention data gathering. The results in Table 1 show that with more interventional data, causal discovery metrics mostly demonstrate a growing trend on TNR, TPR, F1-Score, and accuracy, indicating the positive effect of iterative interventional data gathering.

Table 1: Effect of adding different steps of intervention data

| Set up | TNR | TPR | F1-Score | Accuracy |
|---|---|---|---|---|
| Without intervention | 0.538 | 0.643 | 0.471 | 0.590 |
| 5-step intervention | 0.686 | 0.569 | 0.503 | 0.622 |
| 10-step intervention | 0.695 | 0.628 | 0.528 | **0.656** |
| 20-step intervention | **0.814** | **0.692** | **0.641** | 0.640 |

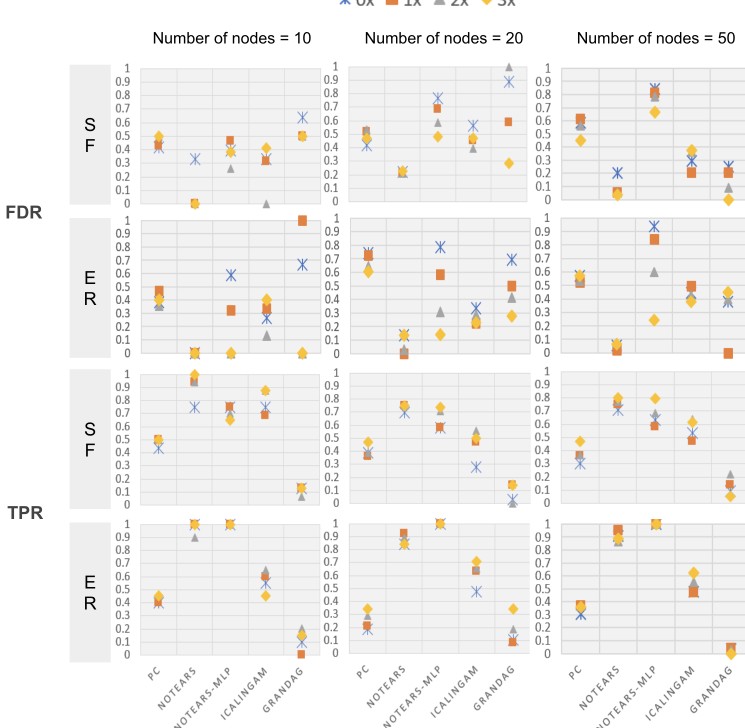

Figure 3: False discovery rate (FDR) and true positive rate (TPR) of causal discovery with different amounts of interventional data using different underlying causal discovery methods in our iterative framework. Columns: varying number of nodes in DAG. The legend indicates how many times of the original data of interventional data is augmented for causal discovery. Results show that interventional data is generally beneficial for causal discovery across various experiment settings.

### 4.3 ITERATIVE RANDOM INTERVENTION ON BASELINE CAUSAL DISCOVERY ALGORITHMS

**Dataset generation.** Following the synthetic data generation approaches in Zheng et al. (2018a), we sample $DAGs$ from two graph generation models: Erdôs-Rényi (ER) and scale-free (SF). We simulate each graph $x_i = f_i\left(\text{pa}(x_i)\right) + z_i$ with gaussian noise $z$ and fixed linear function $f_i$ for fixed sample size $n = 100$. We test on graphs of varying sizes, with number of nodes $d = \{10, 20, 50\}$. We evaluate baselines' performance with two causal discovery metrics: false discovery rate (FDR) and true positive rate (TPR).

**Results on baselines.** We combine the iterative interventional data gathering part with several causal discovery baselines including constrained-based method PC (Kalisch & Bühlman, 2007), gradient-based methods Notears (Zheng et al., 2018b) and its non-linear variant Notears-MLP (Zheng et al., 2020), Grandag (Lachapelle et al., 2019), and function-based method ICALiNGAM (Shimizu et al., 2006) . We test on adding varying numbers of interventional data randomly, as our active intervention policy is not applicable to these baselines, where some of them do not generate a posterior distribution over the edges. Aware of the randomness in experiments, we averaged the results over 5 seeds. From the results demonstrated in Figure 3, more interventional data generally exert positive impacts on the causal discovery performance in terms of FDR and TPR. We noticed that the effect of iteratively augmenting interventional data is not obvious for constrained-based PC algorithm but has a positive impact on Notears, Notears-MLP, Grandag, and ICALiNGAM across various experiment settings.

## 5 CONCLUSION

In this work, we introduced the iterative causal discovery with active interventions. By efficiently collecting interventional data, our algorithm demonstrates performance gains in both causal discovery and downstream prediction tasks. For future work, we plan to test on real-world datasets and design methods to loosen the requirement on simulators.

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
