# OpenReview forum: "Iterative Causal Discovery with Active Interventions"
_ICLR.cc/2022/Workshop/OSC — Submitted to ICLR2022 OSC _

### Official Review · Reviewer_1Vi7 · 2022-03-02
**Review. Briefly: An Oracle-based View on DAG Learning?**

**Rating:** 2
**Confidence:** 3

**Review:**

**1. Summary and Contributions**

The paper considers causal discovery (or induction), that is, learning the underlying causal relations of our data's variables using said data. Specifically, it builds upon previous works that have considered settings in which we might get access to interventional data (on top of our standard, observational data) which leverages the identifiability of the causal structure of interest. The paper proposes an active approach to selecting interventions to be simulated and iterated over for improving our graph estimate. The paper develops the intuitive idea and further provides justification for the practicality of the approach by providing an empirical investigation on synthetic data sets.

**2. Strengths**

The paper has noteworthy strengths, considered one-by-one in the following list (the list is ordered in correspondence to the paper presentation):

* The grand problem of causal discovery is essential to human cognition and thus to science and engineering, and all its instances, and tackling specific sub-problems - as done in this work - is determining for developing next generation learning systems.
* The paper provides a comprehensive context of existing works in causal and more general structure discovery.
* The paper proposes a new, general paradigm to causal discovery that involves access to a simulator (or oracle for that matter). Said paradigm might be re-fined in later iterations through specific inductive biases or prior knowledge potentially rendering it suitable for specific application domains of interest (as motivated in the initial example of reasoning about traffic). [Furthermore, the empirical investigation seems to suggest that settings in which unlikely bla bla]

**3. Weaknesses**

The paper suffers from several disadvantages (ranging in importance from minor to more fundamental) that however IMHO can be improved upon mostly quickly. Thereby, the following list - again one-by-one - aims to provide specific pointers with improvement suggestions if applicable (please note, the list is unordered):

* Consider making the oracle setting more clear. While a white-box simulator is a strong assumption and arguably renders the problem of causal discovery trivial, in this paper the authors seem to consider a black-box simulator (in other words an oracle) and therefore should consider making distinction more clear. To achieve this, the authors might want to also consider examples such as the illustrating example on p.1 concerning traffic.
* The authors should consider removing either Alg.1 or Fig.1 to re-use the extra space for alternate elaborations and improved versions of compact representation of the paradigm (as intended by both Alg.1 and Fig.1), since Alg.1 is arguably simply a more detailed form of presentation of the content in Fig.1 minus the structural information and color coding. An option would be to remove Alg.1, keep Fig.1, incorporate key details that were otherwise only in Alg.1 and add an illustrative, motivating example to the figure (as on p.1 concerning traffic).
* Consider removing or re-iterating the experiment in Sec.4.3. Said experiment does consider only partially the proposed paradigm (since the methods do not allow for a posterior over the edges) therefore it is not corroborating on the proposed paradigm. Furthermore, the key result stated in said experiment (interventional data is better than simply more observational data for recovering the true causal graph) is arguably not of value to the paper since it is (a) to be expected from existing literature and (b) it does not corroborate on the proposed paradigm.
* Consider calling graphical neural networks simply graph neural nets (GNN) to keep consistency with existing literature on GNN.
* Consider using the original invariant causal prediction paper by Peters et al. as reference.
* Consider using an alternate notation for $==$ since it is an arguably atypical notation for graph learning reminiscent of programming language literature.

**3. Correctness, Clarity, and Literature**

No contradictions or any sort of relevant mistake have been detected in the paper. Existing bodies of work are being referenced accordingly at the end of the paper.

**4. Reproducibility, Code Release, and Assumptions**

Sufficient details for reproduction are being provided. Unfortunately, without actual code. All key assumptions for the method are being pointed out explicitly.

---

### Official Review · Reviewer_RmyF · 2022-03-15
**Simple algorithm, unclear how it merges interventional data (possibly wrong), does not cite or compare related work**

**Rating:** 1
**Confidence:** 3

**Review:**

While I think the paper addresses a potentially interesting problem, I have several issues with its current version.

The first issue is that paper proposes a method for intervention design, a well-known problem in causal discovery, which does not cite any of the work on intervention design or any work in which one could learn a causal graph from combinations of interventional and observational data (e.g. the several works mentioned in the introduction of the only related work the authors cite: https://arxiv.org/pdf/2109.02429.pdf).

Moreover, the main contribution of the paper is a very simple algorithm based on the entropy of the posterior.
As has been shown in previous work, this type of approaches (information greedy) is exponentially suboptimal
 in the number of interventions https://papers.nips.cc/paper/2019/file/5ee5605917626676f6a285fa4c10f7b0-Paper.pdf

Another issue I have with the paper, is that it's unclear how interventional data are merged. It seems the authors just add the data from this different distributions without any additional indicator, which would create artifacts if one were to train any causal discovery algorithm on the mixture. (Consider for example two independent variables X and Y, if we collect samples from a two environments in which X and Y are shifted, it might look like a dependence in the mixture, or in other words, you are introducing a latent confounder yourself).

It also confuses me about how one could evaluate methods that were developed for a single distribution (e.g. PC etc) on mixtures of observational and experimental data, especially since methods to deal with different environments exist. For reference you can check a list in Table 4 in https://arxiv.org/pdf/1611.10351.pdf or any of the methods derived from CD-NOD https://arxiv.org/abs/1903.01672.
I'm also confused why the authors don't compare with (Scherrer et al., 2021).

Minor details:

-  The authors also seem to conflate soft and perfect interventions, without mentioning any background or citation on either. These two types of intervention have a quite different treatment in literature and different algorithms
- the main citation for ICP is https://rss.onlinelibrary.wiley.com/doi/full/10.1111/rssb.12167

---

### Decision · Program_Chairs · 2022-03-21

**Decision:**

Reject

**Comment:**

Unfortunately the paper is not ready for presentation at the workshop. I would recommend the authors to take a closer look at reviewer RmyF. While the citation issues are easily resolvable, the other points they raise (suboptimal in the number of interventions, counterexample, experimental setting) need to be addressed. The paper is definitely interesting and I'm looking forward to seeing an improved version sometime in the future!